# pH-Sensitive Alginate/Carboxymethyl Chitosan/Aminated Chitosan Microcapsules for Efficient Encapsulation and Delivery of Diclofenac Sodium

**DOI:** 10.3390/pharmaceutics13030338

**Published:** 2021-03-05

**Authors:** Ahmed M. Omer, Maha S. Ahmed, Gehan M. El-Subruiti, Randa E. Khalifa, Abdelazeem S. Eltaweil

**Affiliations:** 1Polymer Materials Research Department, Advanced Technology and New Materials Research Institute (ATNMRI), City of Scientific Research and Technological Applications (SRTA-City), New Borg El-Arab City, Alexandria 21934, Egypt; rghonim@srtacity.sci.eg; 2Chemistry Department, Faculty of Science, Alexandria University, P.O. Box 426 Ibrahimia, Alexandria 21321, Egypt; maha_said1989@yahoo.com (M.S.A.); gehanmsubruiti@alexu.edu.eg (G.M.E.-S.)

**Keywords:** alginate, carboxymethyl chitosan, aminated chitosan, pH-sensitive microcapsules, polyelectrolyte complex, drug release

## Abstract

To develop an effective pH-sensitive drug carrier, alginate (Alg), carboxymethyl chitosan (CMCs), and aminated chitosan (AmCs) derivatives were employed in this study. A simple ionic gelation technique was employed to formulate Alg-CMCs@AmCs dual polyelectrolyte complexes (PECs) microcapsules as a pH-sensitive carrier for efficient encapsulation and release of diclofenac sodium (DS) drug. The developed microcapsules were characterized by Fourier transform infrared spectroscopy (FT-IR), thermogravimetric analyzer (TGA), and scanning electron microscope (SEM). The results clarified that formation of dual PECs significantly protected Alg microcapsules from rapid disintegration at colon conditions (pH 7.4), and greatly reduced their porosity. In addition, the dual PECs microcapsules can effectively encapsulate 95.4% of DS-drug compared to 86.3 and 68.6% for Alg and Alg-CMCs microcapsules, respectively. Higher DS-release values were achieved in simulated colonic fluid [SCF; pH 7.4] compared to those obtained in simulated gastric fluid [SGF; pH 1.2]. Moreover, the drug burst release was prevented and a sustained DS-release was achieved as the AmCs concentration increased. The results confirmed also that the developed microcapsules were biodegradable in the presence of the lysozyme enzyme. These findings emphasize that the formulated pH-sensitive microcapsules could be applied for the delivery of diclofenac sodium.

## 1. Introduction

Great attention has been given to the development of drug delivery systems (DDSs), which have a direct impact on their potential delivery [1,2]. These systems have been designed to overcome the limitations of conventional therapeutics, since they have the aptitude to enhance the drug solubility, prolong biological activity and improve the therapeutic efficiency as well as extending the drug half-life time [3,4]. Various materials including soluble polymers, microspheres, mesoporous Silica, micro/nanocapsules, lipoproteins, liposomes, micelles, cells ghosts, and emulsions have been well-developed and formulated as effective drug carriers [5,6]. Over the last thirty years, pH-sensitive hydrogels-based marine biopolymers have gained immense interest and proven as promising smart DDSs [7,8]. This interest is due to their beneficial features such as tunable biodegradability, low-cost production, excellent biocompatibility, and non-toxicity [9,10]. Alginate (Alg) among these marine biopolymers is a polyanionic polysaccharide obtained from natural brown seaweeds. It typically consists of 1, 4-linked β-d-mannuronic acid and α-l-guluronic acid monomers [11,12]. Due to its attractive characteristics including high biological safety, non-antigenicity, and biodegradability alginate biopolymer has been employed in several industrial, medical, and pharmaceutical applications [13,14,15]. The interest of the researchers in alginate as a platform for the development of delivery systems has given place to a steady growth of the available literature over the past decade. Therefore, drug carriers-based alginate hydrogel have been suggested as potential delivery systems for various drugs through the gastrointestinal (GI) tract owing to their sensitivity to the physiological pHs [16,17]. Nevertheless, the most common drawbacks of carriers-based alginate are their high porosity which produces drug leakage through their formulation process. Also, their instability at high pH directly leads to the drug burst release at the colon region [18]. Therefore, several physicochemical modifications comprising coating [19,20], grafting [21], and polyelectrolyte complex formation with other cationic polymers have been conducted for alginate to overcome these limitations [22].

Carboxymethyl chitosan (CMCs) is an important water-soluble derivative of chitosan (Cs) biopolymer that is mostly extracted from chitin (the main component for the exoskeleton of crabs, shrimps and lobsters) [23]. In addition to the present cationic amine groups of Cs, the macromolecular structure of CMCs has further anionic carboxylic groups which can provide additional features for Cs biopolymer including ampholytic character and potentials for ample of applications [24,25,26]. Besides, CMCs derivative has been developed to overcome the poor solubility of chitosan in water, which is a major drawback for drug carriers-based chitosan. Compared with other water-soluble chitosan derivatives CMCs is more convenient to be applied in biomedical fields and has been broadly studied for the sustained and targeted delivery of drugs, since it fits the neutral environment of the human body [27,28]. It has been reported that physically crosslinked alginate/N,O-carboxymethyl chitosan hydrogel could be efficiently used as a pH-sensitive system for delivery of proteic drugs to the intestinal tract [29].

Similar to CMCs, amine-functionalized chitosan (aminated chitosan; AmCs) is a fascinating chitosan derivative with extra cationic amine groups [30]. AmCs has been developed to enhance the biological characteristics of the native Cs biopolymer including biocompatibility, hydrophilicity, bioadhesive property, and antibacterial activity [31,32]. Therefore, AmCs has been employed for wound dressing, drug delivery, and water treatment fields [33,34,35]. The authors have developed alginate/chitosan/k-carragennan and alginate coated aminated chitosan microbeads as efficient carriers for the delivery of anticancer (5-Flurouracil) and protein (bovine serum albumin) drugs [16,19].

Taking advantage of polyelectrolyte complex (PEC) formation between cationic chitosan/or chitosan derivatives and anionic alginate biopolymer, an attempt was made in this study to develop pH-sensitive Alg-CMCs@AmCs PECs microcapsules for the delivery of diclofenac sodium (DS; a model of anti-inflammatory drug). Trying to reduce the porosity, impart hydrophilicity, and ameliorate the pH-sensitivity of alginate microcapsules, alginate was combined with CMCs to produce single PEC via the electrostatic interactions between the positively charged –NH_3_^+^ groups of CMCs and the negatively charged –COO^−^ groups of Alg. Next, the prepared Alg-CMCs complex was further modified with a polycation AmCs derivative as a coating layer with a second PEC formation. The second PEC resulted from the strong electrostatic attractions between the extra NH_2_ groups of AmCS and the free -COOH of Alg is expected to enhance the stability of microcapsules, improve the drug encapsulation efficiency, and prevent the drug burst. The developed smart pH-sensitive Alg-CMCs@AmCs microcapsules were characterized using FT-IR, TGA, and SEM analysis tools. Moreover, pH-sensitivity of the developed microcapsules was investigated under the simulated GI-tract conditions. Impacts of CMCs and AmCs concentrations on the drug encapsulation efficiency, swelling, and in vitro drug release profiles were tested. In addition, biodegradability of the formulated microcapsules was also evaluated.

## 2. Material and Methods

### 2.1. Materials

Alginate sodium salt (medium viscosity), O- carboxymethyl chitosan (DS = 34%), diclofenac sodium (DS; assay 99%), calcium chloride (anhydrous; assay 98%) were acquired from Sigma-Aldrich (Taufkirchen, Germany). Aminated chitosan (AmCs; low viscosity) was provided by ATNMRI, SRTA-City (Cairo, Egypt). Acetic acid (assay 98%), hydrochloric acid (assay 37%), di-sodium hydrogen phosphate (anhydrous; ≥99%), sodium di-hydrogen phosphate (≥99%), citric acid (assay 99%), hydrochloric acid, and potassium chloride (assay 99%) were purchased from Aladdin Reagent Co., Ltd. (Shanghai, China).

### 2.2. Preparation of Alg-CMCs Single PEC Microcapsules

Alg-CMCs single PEC microcapsules were prepared using ionic gelation technique [36]. In brief, Alg and CMCs were dissolved separately in distilled water at 60 °C and 25 °C, respectively. Both of Alg and CMCs solutions were mixed together with a final volume of 10 mL under continuous stirring for 1 h at 25 °C. The final Alg: CMCs ratios were 2:0, 1.5:0.5, 1:1, and 0.5:1.5% (*w*/*w*) and coded as S0, S1, S2, and S3, respectively (Table 1). The homogenous Alg-CMCs mixtures were subsequently added dropwise into 200 mL of CaCl_2_ aqueous solution (2%; *w*/*v*) as a gelling medium using an electrostatic pump and a fine syringe needle (3 cm^3^). The distance between the edge of the syringe needle and the gelling solution surface was 10 cm. The formed spherical Alg-CMCs PEC microcapsules were left in the gelling medium for 30 min to harden under constant gentle stirring (25 rpm) at room temperature. After curing, the wet microcapsules were then rinsed three times with distilled water to remove the excess of CaCl_2_, followed by drying overnight at room temperature.

### 2.3. Preparation of Alg-CMCs@AmCs Dual PECs Microcapsules

To generate a second PEC coating layer, the homogenous Alg-CMCs mixture (1:1) was dropped by the same procedure mentioned above into a gently stirred CaCl_2_ solution (2%; *w*/*v*) containing 0.25, 0.5, and 1% (*w*/*v*) of AmCs (prepared by dissolving AmCs in acetic acid (1%; *w*/*v*)). The coating process was conducted at room temperature for 30 min under gentle stirring. After simple filtration, the formulated Alg-CMCs@AmCs dual PECs microcapsules were washed several times with distilled water to eliminate the excess of AmCs and CaCl_2_, and then dried overnight at room temperature. Samples were coded as S4, S5, and S6 for microcapsules coated with 0.25, 0.5, and 1% (*w*/*v*) of AmCs, respectively as presented in Table 1.

### 2.4. DS-Drug Loading Process

To encapsulate DS-drug into the developed microcapsules, DS-drug (0.5%; *w*/*v*) was dissolved in distilled water at 25 °C, and subsequently added to Alg-CMCs mixture to have final polymers: drug ratio of 2:0.5. Thereafter, DS-loaded microcapsules were prepared according to the same procedure described in Section 2.2 and Section 2.3. To determine the drug loading efficiency (LE; %), an accurate quantity of dried microcapsules samples (100 mg) were crushed for easy extraction and soaked in phosphate buffer solution (pH 7.4). The drug extraction process was carried out in a shaking water bath (50 rpm) at 25 °C for 24 h to ensure the completion of DS extraction, followed by filtration step. DS-drug was assayed at 276 nm by UV–spectrophotometer using a standard curve of known DS-drug concentrations with correlation factor of *R* = 0.999. The DS-loading efficiency was estimated according to the following Equation (1):DS-loading efficiency (%) = (*M_1_*/*M_0_*) × 100(1)
where *M_1_* represents the quantity of DS-drug per weighted amount of microcapsules and *M_0_* is the initial quantity of DS-drug.

Figure 1 describes the formulation process of DS-loaded Alg-CMCs@AmCs dual PECs microcapsules as well as digital laboratory images for the freshly prepared wet microcapsules; in addition, the presumptive scheme for the polyelectrolyte complex formation through the electrostatic attractions between Alg, CMCs, and AmCs.

### 2.5. Microcapsules Characterization

The average sizes of the prepared Alg, single, and dual PECs microcapsules were measured in a triplicate using a micrometer screw gauge. Investigation of the functional groups of the native CMCs derivative, native AmCs derivative, and the developed microcapsules was achieved using Fourier transform infrared spectroscopy (FT-IR, Model 8400 S, Shimadzu, Japan). Samples (5–10 mg) were mixed thoroughly with KBr (spectral purity), and the test temperature was set at 25 °C. The absorbance of samples was scanned in the wavenumber range of 500–4000 cm^−1^. Moreover, the thermal properties were inspected by the thermogravimetric analyzer (TGA, Model 50/50H, Shimadzu, Japan). TGA test was performed at a temperature range of 10–800 °C under nitrogen atmosphere with a flow rate of 40 mL/min, and the heating rate was 20 °C/min. Besides, a scanning electron microscope (SEM, Model JSM 6360 LA, Joel, Peabody, MA, USA) was employed to investigate the topographical properties of microcapsules under voltage potential of 15 kV. Prior to SEM examination, the tested samples were placed on aluminum stubs and coated with a thin layer of gold using a sputter coating system.

### 2.6. Swelling Studies

In order to evaluate the swelling profiles of the developed microcapsules, a precise amount of dried samples (0.1 g) was soaked separately at 37 °C in 10 mL of a freshly prepared saline phosphate buffer (pH 7.4) and 0.1 M of hydrochloric acid (pH 1.2). After time intervals (0.5–8 h), the swollen samples were separated, and the excess of water-adhered on the surface of microcapsules was eliminated by blotting them gently between two filter papers. Finally, the swollen samples were weighted in a closed-electronic balance, and the swelling degree (%) was estimated using the following equation:
Swelling degree (%) = (*W_t_* − *W_i_*)/*W_i_* × 100(2)
where *W_i_* represents the initial weight of the dried sample, and *W_t_* is the weight of the swollen sample at time *t*.

### 2.7. DS-Drug Release Studies

The in vitro drug release experiments were conducted in a shaking water bath (50 rpm) at 37 °C. Briefly, 100 mg of DS- loaded microcapsules samples were immersed individually in 10 mL of simulated gastric fluid (SGF; pH 1.2) and simulated colonic fluid (SCF; pH 7.4), which were prepared using fresh buffer solutions. Approximately, one mL of the release medium was removed for assaying after preset time intervals (0.5–8 h) and replaced with a fresh buffer medium with the same pH value, and followed with a simple filtration. A calibration curve of known DS-drug concentrations was performed, and the cumulative released quantity of DS-drug was detected by a UV-spectrophotometer at 276 nm according to the following Equation (2):DS-reléase (%) = (*M_t_* − *M_i_*)/*M_i_* × 100(3)
where *M_t_* represents the released amount of DS-drug at time *t*, and *M_i_* refers to the initial loaded DS amount, respectively.

A further experiment was implemented to evaluate the cumulative released amount of DS-drug through the oral delivery route via simulating the gastrointestinal tract conditions and transit times. Herein, the tested samples were incubated under shaking conditions at 37 °C (50 rpm) in SGF (pH 1.2; 10 mL) solution for 2 h before transferring into SIF (pH 6.8; 10 mL) for additional 3 h, and subsequently, transferred into SCF (pH 7.4; 10 mL) for further 3 h. Then, the same measuring procedure mentioned above was followed for determining the cumulative DS-drug release percentage (%).

The saline buffers used in both the swelling and drug release experiments were prepared using fresh solutions of di-sodium hydrogen phosphate, sodium di-hydrogen phosphate, citric acid, hydrochloric acid, and potassium chloride. The ionic strength was adjusted by sodium chloride solution.

### 2.8. In Vitro Biodegradation Study

Biodegradability of the developed single and dual PECs microcapsules as well as native Alg microcapsules was tested according to the previously reported procedure [37]. The tested samples (0.1 g) were soaked in a mixture of phosphate buffer (2 mL; pH 7.0) and lysozyme solution (0.5 mL) at 37 °C overnight. Lysozyme activity was stopped upon the addition of 3,5-dinitrosalicylic acid (DNS; 1.5 mL) which was used as a reagent for the determination of the reduced sugar. Afterwards, the mixture was boiled at 100 °C for 15 min in a water bath, and then left for cooling. The generated color from reaction of the DNS reagent with the reduced liberated sugar from samples was assayed using a visible-spectrophotometer and the optical density (OD) was estimated at 570 nm.

### 2.9. Statistical Analysis

All experiments were performed in replicates (*n* = 3), and data obtained were presented as means standard deviation (±SD). The significant difference was considered at *p*-value ≤ 0.05.

## 3. Results and Discussion

### 3.1. FT-IR Analysis

IR spectra of native CMCs derivative, native AmCs derivative and the developed microcapsules were obtained to get more information regarding their chemical structures as presented in Figure 2a. The results clarified that the basic characteristics of the chitosan polysaccharide structure (AmCs and CMCs derivatives) were obtained. The observed characteristics comprise broad bands at 3437 and 3447 cm^−1^ in AmCs and CMCs spectra, respectively, which correspond to the stretching vibrations of -NH_2_ and –OH^−^ groups [38,39]. In addition, the IR spectra of AmCs of CMCs derivatives showed absorption bands at 613 and 582 cm^−1^, respectively, which attributed to the C-H deformation of *β*-glycosidic bond. The peaks at 1076–1072 cm^−1^ are assigned to skeletal vibration caused by C-O stretching (unique of its saccharide systems). Likewise, the strong peaks at 1602 and 1420 cm^−1^ in CMCs spectrum are assigned to the asymmetric and symmetric stretching vibration of -COO^−^, respectively [40]. By comparing the IR spectra of Alg (S0) and Alg-CMCs (S2), the observed absorption broad band at 3438 cm^−1^ in case of Alg microcapsules that corresponds to stretching vibration of OH^−^ groups was broadened and shifted to the higher frequency of 3484 cm^−1^ in case of Alg-CMCs. This shifting could be a result of the overlapping between NH_2_ groups of CMCs and OH^−^ groups at the same wavelength. The bands at 1627 and 1430 cm^−1^ (assigned to asymmetric and symmetric -COO^−^ stretching vibration) were shifted to 1639 and 1427 cm^−1^ as a result of formation of intermolecular hydrogen bonds between Alg and CMCs molecules. Furthermore, the peaks at 1035 cm^−1^ and 1032 cm^−1^ in Alg and Alg-CMCs spectra are related to skeletal vibration caused by C-O stretching, and the absorption peaks at 2929–2931 and 460–668 cm^−1^ are associated with the C-H stretching vibration and C-H deformation of *β* glycosidic bond. Similar observations have been noticed by other reported studies [16,18]. On the other hand, FT-IR spectra of Alg-CMCs@AmCs dual PECs microcapsules clarified that all tested samples (S4 and S5) are similar to each other, since there is no observable difference in their spectra owing to the generated Coulomb forces through the second PEC formation. However, the ori.ginal OH^−^ group peak in Alg-CMCs spectra (3484 cm^−1^) was found to be broader at the lower frequencies 3482 and 3479 cm^−1^ in Alg-CMCs@AmCs (S4 and S5) due to the superposition of the stretching vibration of NH_2_ groups in AmCs that overlapped with OH^−^ stretching in Alg and CMCs. Besides, the peaks at 1639 and 1427 cm^−1^ in Alg-CMCs spectra which correspond to asymmetric and symmetric –COO^−^ stretching were moved to 1651 and 1435 cm^−1^ (S5). This shift could be attributed to the formation of a second PEC caused by AmCs. Sharp peaks at 1030–1080 cm^−1^ can also be observed indicating the interaction between NH_3_^+^ of AmCs (outer layer) with -COO^−^ of Alg-CMCs (inner layer), which has been confirmed by other published works [41,42].

### 3.2. TGA Analysis

Figure 2b displays the thermogram curves for native CMCs and AmCs derivatives in addition to Alg (S0), Alg-CMCs, and Alg-CMCs@AmCs microcapsules with elevating temperature up to 800 °C. The results clarified that the thermal profiles for all tested samples took place in several sequential decomposition stages. The first stage was observed with increasing temperature from 25 to 120 °C, with weight losses ranging from 9.51–13.87%. This degradation stage is associated with the release of moisture content from all tested samples [43]. The second stage was observed with rising temperature from 220 to 350 °C and associated with the release of additional bound-water resultant from the interactions with the COOH and NH_2_ groups of microcapsules matrix, decomposition of polymers backbone and destruction of glycosidic bonds (C-O-C) [44].

At the first and second degradation stages, the weight losses in case of Alg-CMCs single PEC and dual Alg-CMCs@AmCs PECs microcapsules were slightly higher than those of Alg microcapsules. These observations could be a result of increasing the hydrophilic nature of Alg microcapsules after mixing and coating with hydrophilic CMCs and AmCs, respectively. The third decomposition stage was noticed with increasing the temperature beyond 350 °C as a result of disintegration of the cyclic compounds, followed by the releasing of CO_2_ molecules from polysaccharides [45]. These decomposition stages were consistent with the reported main thermal decomposition process of alginate and chitosan polysaccharides [46]. With elevating temperature, the thermal stability of the developed microcapsules were significantly improved compared to native CMCs and AmCs derivatives. The compact structure of the developed microcapsules as well as the formation of single and dual PECs reduced the rate of weight loss with rising temperature beyond 350 °C. Therefore, the temperature required for both CMCs and AmCs derivatives to lose their half weights (T_50%_ °C) were recorded at 335.61 and 346.69 °C, respectively (Table 2). On the other hand, Alg microcapsules needed to rise temperature up to 365.39 °C to lose their half weights, while this temperature increased to 377.90 and 384.55 °C in case of Alg-CMCs single PEC and Alg-CMCs@ AmCs dual PECs, respectively. The observed differences between the thermograms confirmed the occurrence of electrostatic interactions that could produce single and dual PECs with diverse thermal characteristics. These results are in agreement with the reported studies in literature [16,42,47]. Finally, it can also be anticipated that the developed PECs microcapsules should be thermally stable at the physiological temperature of the human body.

### 3.3. Morphological Analysis

Through a visual check (Figure 1c), the freshly wet microcapsules were practically spherical with an average diameter approximately of 2.711 ± 0.010–3.891 ± 0.005 mm. Conversely, an extensive decrease in the diameter after drying was noticed and recorded maximum values in the range of 0.484 ± 0.071–0.562 ± 0.090 mm. Moreover, Table 1 clarified that the diameter of wet Alg-CMCs single PEC microcapsules decreased after coating with the AmCs layer as a result of the network shrinking in the acidic conditions. Besides, the topographical properties of microcapsules were examined by a scanning electron microscope (SEM), and the images are depicted in Figure 3. The SEM images of the whole surface indicated that all examined dried microcapsules samples were not completely spherical and constituted polyhedral shapes. Although the drying process of wet microcapsules had been accomplished under mild conditions (i.e., room temperature), the loss of water significantly changed their spherical form. The observed shape deformation is predictable due to the hydrophilic nature of the developed hydrogel microcapsules. In addition, water loss during the dehydration process leads to contraction and collapsing of microcapsules networks. These observations agreed with those obtained by other reports [18,48]. Obviously, the surface of whole Alg microcapsules exhibited a cracking surface with some wrinkles and cavities [49]. Although it became more compact and dense in case of Alg-CMCs due to the generated electrostatic interactions between positively and negatively charged groups resulting in single PEC.

On the other hand, SEM images of the fracture surface of Alg-CMCs@AmCs microcapsules showed a relatively fibrillar and rough surface. It was also noticed that the surface roughness increased with increasing AmCs content in the microcapsules matrix (S4 and S5) due to the interfacial interactions of AmCs chains through the formation of a second PEC [50].

### 3.4. In Vitro Swelling and pH Sensitivity

Definitely, the swelling characteristic is one of the utmost imperative features of the drug carriers-based biopolymers, which directly affects the liquid penetration and the drug release rates. Factually, the swelling phenomenon involves the hydration of the present hydrophilic groups in the microcapsules matrix [51]. The rigid pores among the hydrogel network are filled with the penetrated water molecules from the outer medium, and the swelling occurs accordingly.

The swelling profiles as well as the pH sensitivity of the developed microcapsules were investigated as shown in Figure 4a,b. The results clarified that all tested samples exhibited much higher swelling degrees at pH 7.4 compared with those obtained at pH 1.2, while the rate of increase decreased over time until they reached the equilibrium swelling.

#### 3.4.1. Swelling at pH 1.2

At pH 1.2 (Figure 4a), most of tested Alg (blank; S0) and Alg-CMCs (S1, S2 and S3) single PEC microcapsules maintained their structures and displayed the lower swelling degree values compared with those of dual PECs microcapsules. This is due to the complication of Ca^2+^-Alg which forms a strong core and prevents microcapsules dissociation. In addition, the compact and dense structure of Alg-CMCs single PEC microcapsules could obstruct the swelling. Indeed, the mechanism of pH sensitivity of the developed microcapsules involves the protonation of NH_2_ groups of CMCs and AmCs at the low pH and deprotonation of COOH groups of Alg and CMCs at the high pH [52]. At the strongly acidic conditions (pH 1.2), the hydrated alginate is being altered into a porous and insoluble form (i.e., alginic acid skin), while the overall swelling behavior is mainly controlled by counterions which neutralize NH_3_^+^ groups of CMCs and AmCs. The protonation of the free NH_2_ groups (NH_3_^+^) in the exterior shell of Alg:CMCs@AmCs dual PECs microcapsules prompt greater osmotic pressure and enhance the intermolecular repulsion which leads to microcapsules swelling. Furthermore, the swelling degree was slightly increased from 23 to 32% with increasing AmCs content in the gelling medium from 0.25 to 1% as a result of increasing the hydrophilic NH_2_ groups. These results are in agreement with those reported by the authors previous work [19]. In all cases, the swelling degree values at pH 1.2 did not exceed 32%.

#### 3.4.2. Swelling at pH 7.4

The pH sensitivity of the formulated microcapsules was clearly observed with raising the pH medium from pH 1.2 to pH 7.4 (Figure 4b). The swelling profiles were greatly enhanced at pH 7.4 as a result of deprotonation of carboxylic groups of Alg and CMCs which leads to promoting the electrostatic repulsion between the COO^−^ anions, and consequently, the swelling degree increased [53]. Although Alg microcapsules were rapidly swelled and recorded the highest value of 3050% after 3 h, they were completely destroyed in the swelling medium (saline phosphate buffer; pH 7.4) with increasing time beyond 3 h. This action is due to the higher affinity of Ca^2+^ ions for phosphate ions than alginate, resulting in the dissolution of alginate microcapsules. However, the authors previous works [16,19] have been stated that Alg microbeads disintegrated after 5 h from the initial swelling time. This difference could be mainly attributed to the preparation conditions. On the other hand, combination with CMCs significantly improved the stability of Alg microcapsules in the basic medium (pH 7.4) for 6 h. However, increasing CMCs ratio decreased the swelling rate of Alg-CMCs microcapsules due to increasing the density of the hydrogel network, which delayed the penetration of water molecules into the microcapsules matrix. In addition, increasing CMCs ratio creates multi-hydrogen bonds as well as generates a strong PEC between the negatively charged COO^−^ of Alg and the positively charged NH_3_^+^ groups of CMCs (Figure 1a). Thus, some of the free hydrophilic groups were consumed through the electrostatic interactions, and the swelling degree decreased consequently.

Besides, the existence of the AmCs coated layer in the outer shell of Alg:CMCs@AmCs microcapsules greatly prevented the dissolution of Alg microcapsules in the swelling medium (pH 7.4) and overcame their high porosity. Where, the swelling time was extended to 8 h without dissolution of microcapsules, while the swelling degree values were decreased as AmCs concentration increased. Reducing the swelling degree with increasing AmCs concentration from 0.25 to 0.5% could be a result of deprotonation of some -NH_3_^+^ groups of AmCs which become a hydrophobic layer [18]. At the same time, the other hydrophilic NH_2_ groups could be involved in the electrostatic interactions with the free negative charges of Alg resulting in a more significant fraction of a second PEC formation. Accordingly, these groups did not contribute to holding the water molecules within the network of microcapsules, which resulted in a decrease in their swelling degrees. However, further increasing AmCs concentration up to 1% was not effective and the swelling degree variation was not considerable, since the osmotic pressure responsible for the network swelling is commonly motivated by the counterions that neutralized the carboxylic groups in the microcapsules matrix.

### 3.5. Evaluation of DS-Loading

To accomplish the anticipated therapeutic effect of the drug dosage, the oral drug carriers require substantial encapsulation efficiency without obstructing the biological action of drugs. The main objective of this study is to improve the drug encapsulation efficiency of microcapsules-based alginate. Figure 5, revealed that Alg microcapsules (S0) recorded the lowest loading value of 68.6% due to their high porosity. The DS-loading efficiency was greatly improved after the formation of single PEC, since it increased with increasing CMCs ratio and reached a maximum value of 86.3% using Alg-CMCs (1:1) single PEC microcapsules. In addition, Alg:CMCs@AmCs dual PECs microcapsules (S4, S5 and S6) displayed the highest values compared to those obtained by neat Alg and Alg:CMCs single. The DS-loading efficiency was increased from 90.23 to 95.45% with increasing AmCS concentration from 0.25% (S4) to 0.5% (S5) as a result of increasing viscosity of the gelling medium (CaCl_2_/AmCs). Furthermore, increasing AmCs concentration would generate more electrostatic attractions between the free positively charged NH_2_ groups on the surface of microcapsules and the negatively charged DS-molecules. These in turn prevent the leakage of DS molecules through the formulation of coated microcapsules. In contrast, raising the AmCs concentration in the gelling medium from up to 1% would produce a highly viscous solution that adversely affects the dispersion of DS-molecules through the microcapsules matrix, resulting in a slight decrease in the encapsulation efficiency. A comparison of the obtained DS-loading efficiencies with those obtained by other reported studies using carriers-based alginate biopolymer is presented in Table 3 [54,55,56,57,58,59,60]. The results clarified that the developed single and dual PECs microcapsules exhibited better loading efficiencies, confirming the successful formulation process.

According to the swelling profiles as well as the DS-loading efficiency values, samples of S2 (Alg-CMCs [1:1]), S4 (Alg:CMCs [1:1]@AmCs [0.25%AmCs]), and S5 (Alg:CMCs [1:1]@AmCs[0.5%AmCs]) were selected in addition to S0 (Alg) sample (blank) for the subsequent drug release and biodegradation studies.

### 3.6. In Vitro DS-Drug Release

Indeed, the drug release mechanism is mostly controlled by the network swelling of the hydrogel microcapsules at the initial stage. The release profiles of DS-drug were investigated in simulated gastric fluid (SGF; pH 1.2) and simulated colonic fluid (SCF; pH 7.4) as shown in Figure 6a,b. The results portrayed that pH-sensitivity of alginate, carboxylated chitosan and aminated chitosan can be exploited to customize the DS-drug release profiles. By comparing the drug release profiles, it was evident that the drug release (%) in SGF (pH 1.2) was by far lower than that of SCF (pH 7.4), which complies with the swelling profiles at the same pH as mentioned previously. The results revealed that both of Alg and Alg-CMCs did not show a significant DS-drug release in SGF pH 1.2 microcapsules, since maximum release values did not exceed 7 and 10% after 8 h for Alg and Alg-CMCs microcapsules, respectively. The limited drug release in SGF could be a result of lower swelling degrees of tested microcapsules at the same pH 1.2. In addition, the stable structure of both Alg and Alg-CMC single PEC microcapsules would prevent the swelling of hydrogel networks and consequently obstruct the DS- drug release in SGF. However, a slight increase in the drug release (%) was noticed in case of Alg-CMC@AmCs dual PECs microcapsules (S4 and S5) due to protonation of some free NH_2_ groups in the outer layer. This in turn, facilitates the solution diffusion [61], and DS-drug would be released accordingly.

In contrast, the DS-drug release profile was obviously improved with rising pH of the swelling medium to pH 7.4 owing to the amphoteric character of alginate and chitosan derivatives at the high pH. For Alg microcapsules (S0), most of DS- drug was released (97%) within 3 h and then, the drug burst release process took place as a result of rapid dissociation of Alg microcapsules in the saline phosphate buffer (pH 7.4) [62]. Besides, the DS-drug release value in case of Alg-CMCs single PEC microcapsules (S2) was lower compared with native Alg microcapsules, while the release time was extended up to 6 h with a maximum release value of 93%. These observations could be ascribed to increasing the network density of hydrogel microcapsules, which delays both swelling and drug release rates.

On the other hand, the coating of microcapsules by AmCs layer significantly extended the release rate of DS-drug up to 8 h, and protected the Alg-CMCs@AmCs microcapsules from the fast dissociation AmCs in pH 7.4. In addition, the low solubility of AmCs at pH 7.4 as well as the deprotonation of -NH_3_ groups specifically in the outer shell of microcapsules led to decreasing the chemical potential of the whole network for water uptake. Therefore, the release (%) decreased consequently compared to Alg and Alg-CMCs single PEC microcapsules. The release data indicated that increasing AmCs concentration in the gelling medium from 0.25% (S4) to 0.5% (S5) restricted the entrance of water molecules into the hydrogel network, and thus, a decrease in DS-drug release (%) from 88% to 74% was observed.

### 3.7. Cumulative DS-Drug Release in Simulated GI-Tract

It is well known that the overall gastrointestinal (GI) transit time for the drug dose could vary from patient to another depending on their physiological conditions [63]. In most healthy cases, the oral drug carrier goes directly to the stomach site and stays for a time period ranging from 2 to 4 h, then it moves to the small intestine and stays there for 3 h before finally reaching the colon. Herein, the cumulative DS-drug release profiles were evaluated under the simulated gastrointestinal tract conditions and the results are displayed in Figure 7a. Due to the good stability of Alg core matrix as well as the dominant effect of COOH groups of Alg and CMCs at the simulated gastric fluid (SGF; pH 1.2), a tiniest cumulative amount of DS-drug (not more than 10%) was released within 2 h from Alg (S0) and Alg-CMCs (S2) single PEC microcapsules. Whereas the swellability of AmCs in the outer shell of Alg-CMCs@AmCs microcapsules causes a slight increase in the released amount of DS-drug, while the overall release amount of DS did not exceed 18% in SGF. Once samples transferred into the simulated intestinal fluid [SIF; pH 6.8], a sharp increase in the drug release rate was noticed within 3 h for all microcapsules. These results could be explained by the progressive ionization of COOH groups of Alg and CMCs, since the hydrogel network being swelled, making it much easier for the SD-drug to release through the microcapsule barrier and enter the intestine. Besides, a complete erosion occurred for Alg microcapsules (S0) after 1 h from transferring into the simulated colonic fluid [SCF; pH 7.4], since most of the enveloped DS-drug was released quickly from Alg microcapsules (S0) due to their high porosity. In contrast, both single and dual PECs microcapsules displayed a continuous drug release in SCF for 3 h with a slow release rate. Furthermore, increasing of the AmCs concentration from 0.25% (S4) to 0.5% (S5) decreased the cumulative drug release (%) from 83 to 74%. These observations could be a result of the successful formation of dual PECs via coating with AmCs external layer, which controlled the release of DS-drug in SCF pH 7.4 via the steric hindrance. Also, more rough, dense, and crusty surface could be produced, which is expected to delay the network swelling. Accordingly, the encapsulated DS-drug further diffused through a longer distance to reach the surface resulting in a sustained drug release.

### 3.8. Biodegradability

One of the most substantial characteristics for drug carriers-based natural polymers is their ability to biodegrade under the physiological conditions after the completion of the drug release process [64]. Figure 7b shows the biodegradability profiles of the developed microcapsules in presence of the lysozyme enzyme. The results clarified that all tested samples were biodegraded due to the biodegradability nature of chitosan and alginate-based marine sources. It is well-known that the glycosidic bonds could be hydrolyzed by the lysozyme enzyme, resulting in a biodegradation of the polysaccharide structure [65,66]. Furthermore, the hydrophilic groups (OH, COOH, and NH_2_) in the microcapsules matrix might enhance the enzymatic activity, which would potentially allow for adsorption of lysozyme enzyme. Therefore, the biodegradation rate was increased with increasing AmCs content as a result of increasing the enzyme-adsorbing groups, which positively affected the biodegradation process. These results are consistent with the authors’ previous works. It has been stated that microbeads, nanoparticles, and membranes based-chitosan biopolymer displayed decent biodegradability [16,19,38,67].

## 4. Conclusions

In this study, Alg-CMCs@AmCs dual PECs microcapsules were developed in order to improve the drug encapsulation efficiency as well as to overcome the fast drug burst release from Alg microcapsules at high pH. Several instrumental tools including FT-IR, TGA, and SEM were used for characterizing microcapsules. pH sensitivity of the developed microcapsules was inspected through studying their swelling profiles at pH 1.2 and pH 7.4. The formulated dual PECs microcapsules exposed the highest DS-loading efficiency value (95.4%) compared to that obtained for native Alg microcapsules. The results signified that formation of PECs greatly improved the stability of Alg microcapsules and triggered a self-sustained DS-release profile in simulated colonic fluid. Moreover, with increasing AmCs concentration from 0.25% (S4) to 0.5% (S5) the cumulative DS-release (%) was decreased from 83 to 74% after 8 h (as overall GI-transit time). Besides, all developed microcapsules were biodegraded which is beneficial for their potential use as pH-responsive drug delivery systems.

## Figures and Tables

**Figure 1 pharmaceutics-13-00338-f001:**
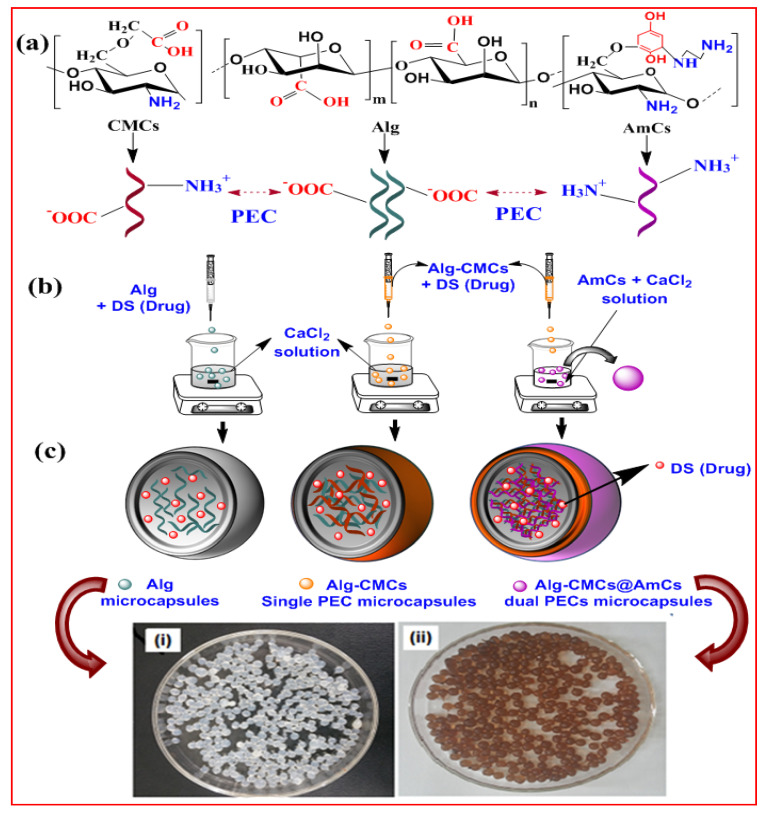
(**a**) proposed scheme for the polyelectrolyte complex formation between alginate (Alg), carboxymethyl chitosan (CMCs), and aminated chitosan (AmCs), (**b**) a schematic diagram for the formulation of diclofenac sodium (DS)-loaded Alg, Alg-CMCs, and Alg-CMCs@AmCs dual polyelectrolyte complexes (PECs) microcapsules and (**c**) digital laboratory image for freshly (**i**) Alg and (**ii**) Alg-CMCs@AmCs wet microcapsules.

**Figure 2 pharmaceutics-13-00338-f002:**
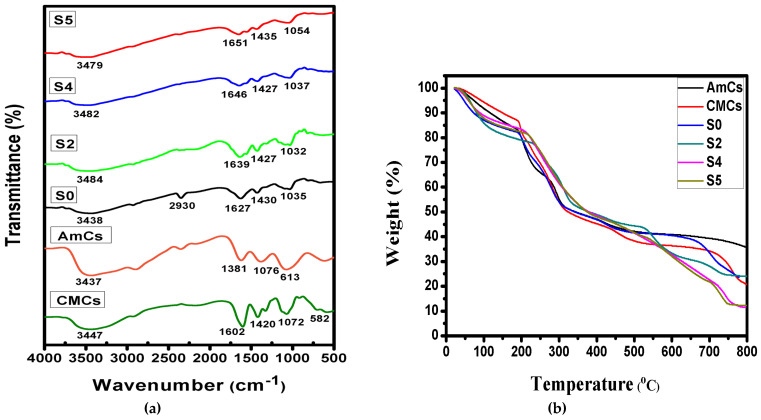
(**a**) FTIR spectra and (**b**) TGA thermograms of native AmCs, CMCs derivatives, Alg (S0), Alg-CMCs (S2), and Alg-CMCs@AmCs (S4 and S5) microcapsules.

**Figure 3 pharmaceutics-13-00338-f003:**
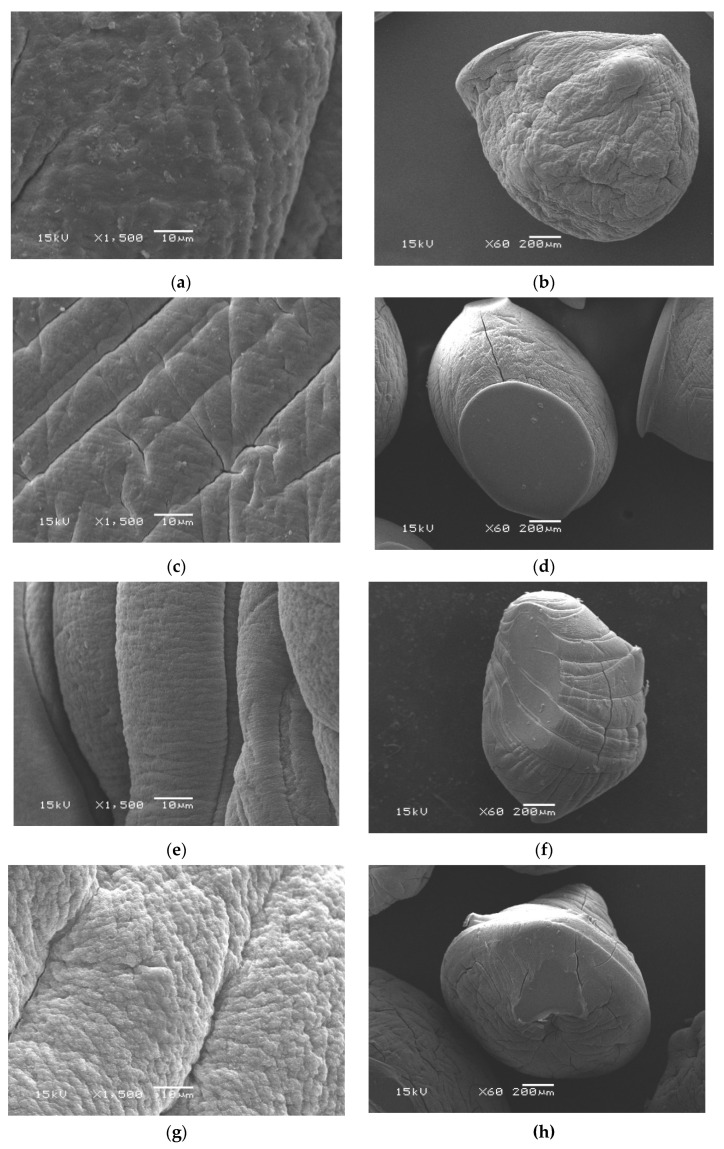
SEM images of fracture and whole surface of (**a**,**b**) Alg (S0), (**c**,**d**) Alg-CMCs (S2), (**e**,**f**) Alg-CMCs@AmCs (S4) and (**g**,**h**) Alg-CMCs@AmCs (S5).

**Figure 4 pharmaceutics-13-00338-f004:**
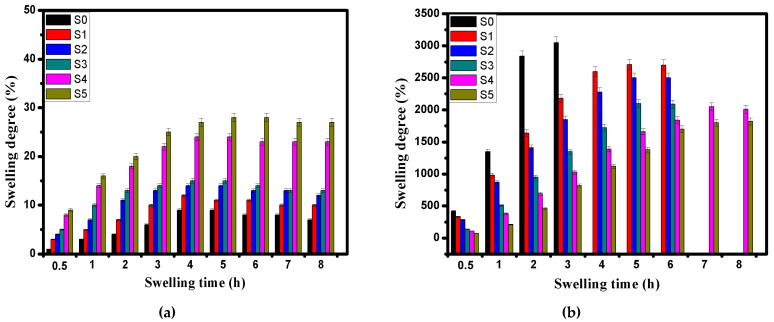
Swelling profiles for Alg (S0), Alg-CMCs single PEC (S1, S2, S3) and Alg-CMCs@AmCs dual PECs (S4, S5) microcapsules at (**a**) pH1.2 and (**b**) pH 7.4. Values were presented as means ± S.D (*n* = 3), and significant data were determined at *p*-value *≤ 0.05*.

**Figure 5 pharmaceutics-13-00338-f005:**
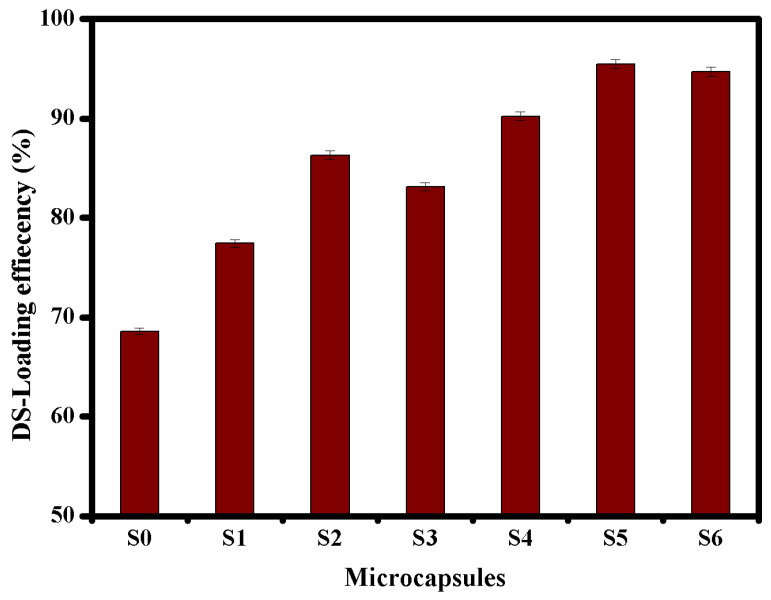
DS-loading efficiency values for Alg (S0), Alg-CMCs single PEC (S1-S3) and Alg-CMCs@AmCs dual PECs (S4–S6) microcapsules. Values were presented as means ± S.D (*n* = 3), and significant data were determined at *p*-value ≤ 0.05.

**Figure 6 pharmaceutics-13-00338-f006:**
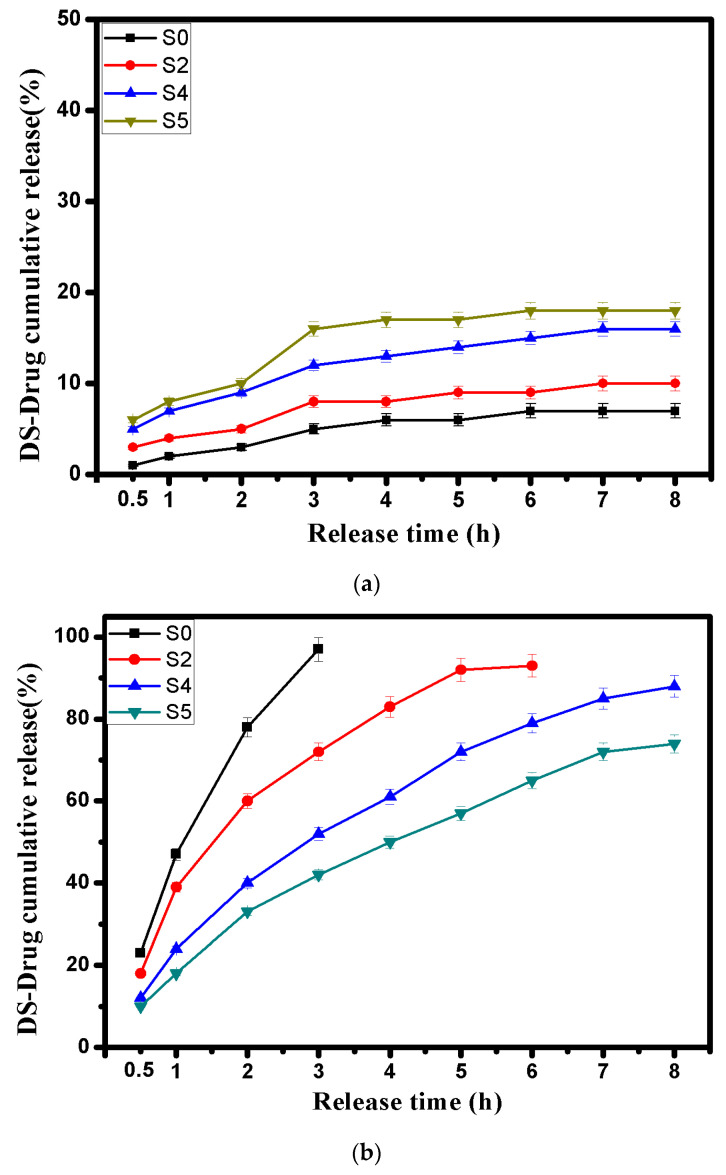
DS-drug release profiles for Alg (S0), Alg-CMCs single PEC (S2) and Alg-CMCs@AmCs dual PECs (S4 and S5) microcapsules at (**a**) pH1.2 and (**b**) pH 7.4. Values were presented as means ± S.D (*n* = 3), and significant data were determined at *p*-value ≤ 0.05.

**Figure 7 pharmaceutics-13-00338-f007:**
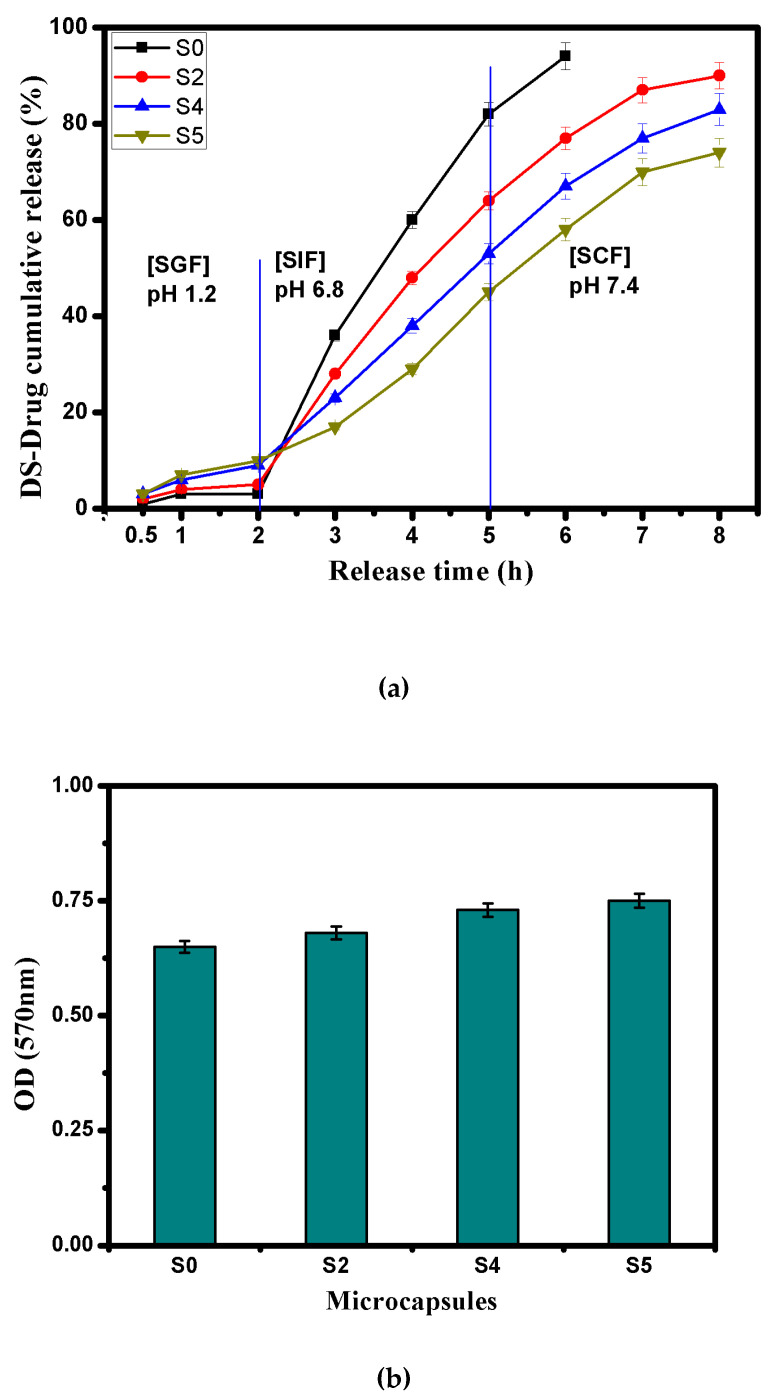
(**a**) Cumulative DS-drug release profiles at simulated GI-tract conditions and (**b**) biodegradation profiles for (S0) Alg, (S2) Alg-CMCs, and (S4 and S5) Alg-CMCs@AmCs microcapsules. Values were presented as means ± S.D (*n* = 3), and significant data were determined at *p*-value ≤ 0.05.

**Table 1 pharmaceutics-13-00338-t001:** Compositions and average diameter of the developed microcapsules.

Samples	Compositions	Diameter (mm)	
		Wet Microcapsules	Dried Microcapsules
S0	Alg-CMCs (2:0)	3891 ± 0.005 ^a^	0.562 ± 0.090
S1	Alg-CMCs (1.5:0.5)	3821 ± 0.012	0.522 ± 0.041
S2	Alg-CMCs (1:1)	3684 ± 0.020	0.5421 ± 0.082
S3	Alg-CMCs (0.5:1.5)	3735 ± 0.070	0.515 ± 0.120
S4	Alg-CMCs (1:1)@ 0.25% AmCs	3133 ± 0.081	0.498 ± 0.061
S5	Alg-CMCs (1:1)@ 0.5% AmCs	2915 ± 0.050	0.491 ± 0.020
S6	Alg-CMCs (1:1)@ 1%AmCs	2711 ± 0.010	0.484 ± 0.071

^a^ Values are expressed as mean ± S.D (*n* = 3), and significant data were determined at *p*-value ≤ 0.05.

**Table 2 pharmaceutics-13-00338-t002:** TGA data for CMCs, AmCs, Alg (S0), Alg-CMCs (S2), and Alg-CMCs@AmCs (S4 and S5) microcapsules.

Sample	Weight Loss (%)		T_50%_°C
	25–120 °C	220–350 °C	
CMCs	9.51	29.11	335.61
AmCs	11.12	22.27	346.69
Alg (S0)	11.7	24.222	365.39
S2	12.38	26.240	377.90
S4	13.67	27.576	384.55
S5	13.87	27.509	383.09

**Table 3 pharmaceutics-13-00338-t003:** Comparison of DS-loading efficiency of Alg-CMCs@AmCs microcapsules with other reported drug delivery systems-based alginate biopolymer.

Drug Delivery System	DS-Loading Efficiency (%)	Ref.
Alginate microspheres	49.97	[54]
Sodium alginate beads	59.88–74.44	[55]
Alginate/carboxymethyl cellulose mono and bilayer films	57.5–77.3	[56]
Floating Alginate Beads	75–88.9	[57]
Sodium alginate-Pectin microbeads	70.40–88.20	[58]
Alginate grafted chitosan microcapsules	89.0	[59]
Alginate-okra gum blend beads	89.27	[60]
Alg-CMCs@AmCs microcapsules	95.45	this study

## Data Availability

The data presented in this study are available on request from the corresponding author.

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
