# Peer review of "pH-Sensitive Alginate/Carboxymethyl Chitosan/Aminated Chitosan Microcapsules for Efficient Encapsulation and Delivery of Diclofenac Sodium"

_pharmaceutics, 2021, doi:10.3390/pharmaceutics13030338_

Round 1

Reviewer 1 Report

Table 1 Average diameter is given. However also the PDI and standard deviation should be provided.

TGA results are pure speculation without comparison to literature. Why are not all 6 samples shown? For FTIR and TGA?

Table 2 it is not known if the differences are significant.

It is not clear how the SEM are relevant. 1. SEM is conducted under vacuum – and the CMCs will change in shape due to the vacuum – unless the CMCs were frozen before – however there is no explanation of the procedure.

Figure 4 – statistical significance should be included.

The biodegradation study is not clear. Why was it done in the first place? It should be obvious that the material is biodegradable. No comparisons are done with literature.

Author Response

We would like to thank reviewer for the valuable comments, we have modified the manuscript and answered all comments according to the reviewer suggestions; we have improved the introduction section and provided sufficient background regarding the developed microcapsules. Experimental, results/discussion and conclusion section were also improved. All changes are highlighted in yellow.

1-Table 1 Average diameter is given. However also the PDI and standard deviation should be provided.

Response: We have provided and mentioned the standard deviation for all measured microcapsules diameter (Table 1), however we didn’t provide the polydispersity index (PDI) values, since we used a simple a micrometer screw gauge for measuring the particle size (micro scale) and it doesn't include the option for PDI measuring, we didn’t use Particle size analyzer. The measurements were performed in triplicate and the average mean values were provided, and values are expressed as means standard deviation (±SD; n=3).

2-TGA results are pure speculation without comparison to literature. Why are not all 6 samples shown? For FTIR and TGA?

Response: We have provided more discussion compared with literature, in addition TGA for native AmCs and CMCs were added (according to the reviewer 4). We have provided the TGA analysis for the selected samples depending on the best results obtained from swelling and drug encapsulation experiments. Thus, the pure alginate (S0), one ratio of S2 [Alg-CMCs (1:1)], and two ration for Alg-CMCs (1:1)@ AmCs [S4 and S5]. In addition, the cost of TGA test is expensive, since we have no fund to cover the analysis, we paid from our own money; as a result, we selected the samples according to the optimum and best results.

3-Table 2 it is not known if the differences are significant.

Response: In fact, data presented in Table 2 were extracted from the original data obtained by TGA. By using the start weight sample (100%) and the residual sample weight (%) after elevating temperature up to 800 °C. The differences could not be clearly observed in case of tested microcapsules because they have similar compositions, which have the same polysaccharides nature. However, with elevating temperature, the significant difference was observed compared to the thermal stability of neat AmCs and CMCs derivatives with that obtained in case of tested microcapsules due to the modification and formulation processes. We have improved the TGA discussion to be clear for readers.

  1. It is not clear how the SEM are relevant. 1. SEM is conducted under vacuum – and the CMCs will change in shape due to the vacuum – unless the CMCs were frozen before – however there is no explanation of the procedure.

Response: The topographical properties of microcapsules were examined by a scanning electron microscope (SEM), and the images were depicted in Figure 3. The SEM images of the whole surface indicated that all examined dried microcapsules samples were not completely spherical and constituted of polyhedral shapes. Although the drying process of wet microcapsules had been accomplished under mild conditions (i.e. room temperature), the loss of water significantly changed their spherical form. The observed shape deformation is predictable due to the hydrophilic nature of the developed hydrogel microcapsules. In addition, water loss during the through the dehydration process leads to contraction and collapsing of microcapsules networks. These observations agreed with those obtained by other reports [18, 48]. Obviously, the surface of whole Alg microcapsules exhibited a cracking surface with some wrinkles and cavities [49]. Although it became more compact and dense in case of Alg-CMCs due to the generated electrostatic interactions between positively and negatively charged groups resulting in single PEC.

On the other hand, SEM images of the fracture surface of Alg-CMCs@AmCs microcapsules showed a relatively fibrillar and rough surface. It was also noticed that the surface roughness increased with increasing AmCs content in the microcapsules matrix (S4 and S5) due to the interfacial interactions of AmCs chains through the formation of a second PEC [50].

5- Figure 4 – statistical significance should be included.

Response: We have added the following sentence to the caption figure"Values were presented as means ±S.D (n=3), and significant data were determined at P-value ≤ 0.05".

6-The biodegradation study is not clear. Why was it done in the first place? It should be obvious that the material is biodegradable. No comparisons are done with literature.

Response: We have modified the discussion of biodegradability study and supported with new references. In fact, several routs have been reported for investigation the biodegradability of alginate and chitosan biopolymers. Some reports examined the biodegradation profiles via estimation the weight loss in several subsequent days (Kenawy, et al., 2019;  Lončarević, et al., 2017; Lim, et al., 2008), while others (Sun, et al., 2019; Omer, et al., 20121; Omer, et al., 2016) studies have been applied the same simple adopted method in the current study. We have performed this test as an indicator for the possible biodegradation of the developed microcapsules, which already have biodegradation nature resultant from origin nature of chitosan and alginate polysaccharides. The produced color from the reaction of the reduced liberated sugar (refers the weight loss) from microcapsules with the DNS-reagent was analyzed as a function of the optical density (OD) at a wavelength of 570 nm using a visible-spectrophotometer.

It is well known that the glycosidic bonds could be hydrolyzed by the lysozyme enzyme, resulting in a biodegradation of the polysaccharide structure [65, 66]. Furthermore, the hydrophilic groups (OH, COOH and NH2) in the microcapsules matrix might be enhanced the enzymatic activity, which would potentially allow for adsorption of lysozyme enzyme. Therefore, the biodegradation rate was increased with increasing AmCs content as a result of increasing the enzyme-adsorbing groups, which positively affected the biodegradation process. These results are consistent with the authors previous works. It has been stated that microbeads, nanoparticles, and membranes based-chitosan biopolymer displayed decent biodegradability [16, 19, 38, 67].

Reviewer 2 Report

Alginate microbeads have been prepared in previous work. While in the current work the microbeads disintegrate after 3h at pH 7.4 (Swelling and drug release experiments) in previous works (Ref. 15) the microbeads do not seem to behave in the same way. What is the difference?

Authors may want to justify differences or similarities found between Alg-CMCs-AmCs (this work) and Alg-AmCs (Ref. 15) transport systems.

Review the following points:

Line 29. “colon” Colon (Keywords)

Line 45. “-d- , and -l-“ should be -D- and -L-

Lines 134-136. The procedure for determining M1 should be more detailed: if a calibration line is used, etc. Reference 31 may not be necessary.

Line 146. In general, experimental details concerning IR, TGA, SEM should be given in with more detailed ... only the equipment used is indicated.

Line 160. Reference 32 is not necessary.

Line 171. Same as said for lines 134-136. Reference 33 is not necessary.

Line 214. “3484” should be 3482

Line 218. “(S4)” should be (S5).

Line 227-228. “at ambient temperature (around 120 oC)”?!

Lines 238-240. Reference 35 does not refer to what is mentioned in these lines. There is no thermal study.

Line 320. “delayd”

Figure 2a. It is not of good quality. It is not very useful as it is difficult to appreciate what is described in the text.

Figure 2b. On the ordinate axis it is Weight (%), not Weight loss%

Figure 4. Swelling profiles for Alg (S0), .. also include (S1) and (S3) in Figure Caption

Figure 5. The structures are not well drawn: Alg: HO-C3 equatorial, C6 send backwards, do not include the two oxygens inside the bracket. AmCs the subscripts of amino, methylene. Its quality is not good.

Table 2. “Ambient 0-120 oC”. It should be Room temperature-120 or 0-120 oC (What is the temperature range used in the TGA?)

References. In some references the names of the journals are in lower case and in others in capital letters. For example: International journal of biological macromolecules vs Journal of Controlled Release.

Line 520, 561 and 572. Missing data for its correct location.

Line 562, Reference 40. The journal is: International journal of Pharmacy and Pharmaceutical Sciences.

Line 581. subscript in ZnSO4.

Author Response

We would like to thank reviewer for the valuable comments, we have modified, corrected and improved the manuscript and answered all comments according to the reviewer suggestions

1-Alginate microbeads have been prepared in previous work. While in the current work the microbeads disintegrate after 3h at pH 7.4 (Swelling and drug release experiments) in previous works (Ref. 15) the microbeads do not seem to behave in the same way. What is the difference?

Response: Thanks to the reviewer for his important scientific notes. In fact, the disintegration of Alg microbeads (after 3 h in this study and 5h in the previous study) in addition to the difference between the swelling/release profile of Alg microbeads in the previous study and the current study is mainly attributed to the preparation conditions.

-In this study: the formed wet Alg microcapsules were dried overnight at room temperature, while in the previous study they have been left  for drying at 40◦C overnight. In addition, the average diameter of the formed microbeads in the current study was 0.56mm, while it recorded 0.70mm in the previous study.

 In this study; the Alg microcapsules were formed by using fine syringe needle (3 cm3), 200 mL of CaCl2, gelling time 30 min under a constant gentle stirring (25 rpm) at room temperature), while the conditions in the previous study were: fine syringe needle (5 cm3, diameter 0.45 mm), 100 mL of CaCl2.

In this study: the used calcium chloride (anhydrous 90%) and obtained from Sigma-Aldrich (Germany), while in the previous study, it was (anhydrous 98%) was purchased from Fisher Scientific (Fairlawn, USA).

Besides; the difference in the buffer compositions source might be cause the difference in the Alg swelling and release profiles

2- Authors may want to justify differences or similarities found between Alg-CMCs-AmCs (this work) and Alg-AmCs (Ref. 15) transport systems.

Response: We have mentioned the differences and similarities found between Alg-CMCs-AmCs (this work) and Alg-AmCs (Ref. 15) transport systems. Ref. 15 changed to be Ref. 19.

3- Review the following points:

-Line 29. “colon” Colon (Keywords)

Response: We have changed it to Colon.

-Line 45. “-d- , and -l-“ should be -D- and -L-

Response: We have changed it

Lines 134-136. The procedure for determining M1 should be more detailed: if a calibration line is used, etc.

Response: We have mentioned that a standard curve of known DS-drug concentrations with correlation factor of R=0.999 was performed.

Reference 31 may not be necessary.

Response: We have deleted it.

Line 146. In general, experimental details concerning IR, TGA, SEM should be given in with more detailed ... only the equipment used is indicated.

Response: We have provided more details concerning IR, TGA, SEM equipment and analysis conditions as follow: " Investigation of the functional groups of the native CMCs derivative, native AmCs derivative and the developed microcapsules was achieved using Fourier transform infrared spectroscopy (FT-IR, Model 8400 S, Shimadzu, Japan). Samples (5–10 mg) were mixed thoroughly with KBr (spectral purity), and the test temperature was set at 25 °C. The absorbance of samples was scanned in the range from 4000 to 500 cm–1. Moreover, the thermal properties were inspected by the thermogravimetric analyzer (TGA, Model 50/50H, Shimadzu, Japan). TGA test was performed at a temperature range of 10–800 °C under nitrogen atmosphere with a flow rate of 40mL/min, and the heating rate was 20 °C/min. Besides, a scanning electron microscope (SEM, Model JSM 6360 LA, Joel, USA) was employed to investigate the topographical properties of microcapsules under voltage potential of 15 kV. Prior to SEM examination, the tested samples were placed on aluminum stubs and coated with a thin layer of gold using a sputter coating system".

Line 160. Reference 32 is not necessary.

Response: We have deleted it.

Line 171. Same as said for lines 134-136.

Response: We have mentioned in the experimental section that a standard curve of known DS-drug concentrations was performed to estimate its concentration along with time.

 Reference 33 is not necessary.

Response: We have deleted it.

Line 214. “3484” should be 3482

Response: We have corrected it

 Line 218. “(S4)” should be (S5).

Response: We have corrected it

Line 227-228. “at ambient temperature (around 120 oC)”?!

Response: We have changed it to 25-120 oC

Lines 238-240. Reference 35 does not refer to what is mentioned in these lines. There is no thermal study.

Response: We have deleted it and add 7 references relevant to the thermal study.

Line 320. “delayd”

Response: We have corrected it

Figure 2a. It is not of good quality. It is not very useful as it is difficult to appreciate what is described in the text.

Response: We have improved the quality of Figure 2a and provided more discussion. In addition, native AmCs and CMCs derivatives were also added and their characteristics were also described according to the comment of reviewer 4.

Figure 2b. On the ordinate axis it is Weight (%), not Weight loss%

Response: We have corrected it, and the fig. number changed to Figure 3.

Figure 4. Swelling profiles for Alg (S0), .. also include (S1) and (S3) in Figure Caption

Response: We have corrected it and included (S1) and (S3) in Figure Caption, and the fig. number changed to Figure 5.

Figure 5. The structures are not well drawn: Alg: HO-C3 equatorial, C6 send backwards, do not include the two oxygens inside the bracket. AmCs the subscripts of amino, methylene. Its quality is not good.

Response: We have corrected it, and the figure was merged with figure 1 according to the comment of reviewer 4.

Table 2. “Ambient 0-120 oC”. It should be Room temperature-120 or 0-120 oC (What is the temperature range used in the TGA?)

Response: We have changed it to 25-120 oC. The temperature range wan from 25 to 800 oC.

References. In some references the names of the journals are in lower case and in others in capital letters. For example: International journal of biological macromolecules vs Journal of Controlled Release.

Response: We have corrected and unified all reference in one reference style format

Line 520, 561 and 572. Missing data for its correct location.

Response: We have corrected and provided the missed data.

Line 562, Reference 40. The journal is: International journal of Pharmacy and Pharmaceutical Sciences.

 Response: We have corrected it

Line 581. subscript in ZnSO4.

Response: We have corrected it

Reviewer 3 Report

Omer and coauthors reported a DDS system based on alginate and chitosan derivatives, with analysis on the encapsulation efficiency and pH-response release efficiency. Comparing this work with many previous works on the microcapsules based on alginate or chitosan, it is hard for the reviewer to find any new contribution of this work to the research field. Therefore, it is very hard for the reviewer to provide a more positive recommendation on the current version.

Besides, there are major parts for revision:

1, in the introduction, the innovations are not clearly demonstrated, especially, the progress in this field is not well summarized with the existing problems. And many works on alginate microcapsules are not discussed, such as:

[1] Preparation of Ca-alginate-whey protein isolate microcapsules for protection and delivery of L. bulgaricus and L. paracasei, International Journal of Biological Macromolecules

Volume 163, 15 November 2020, Pages 1361-1368

[2] Development, characterization and in vitro antioxidant activity of chitosan-coated alginate microcapsules entrapping Viola odorata Linn. Extract, International Journal of Biological Macromolecules, Volume 163, 15 November 2020, Pages 44-54

[3] Physically crosslinked alginate/N,O-carboxymethyl chitosan hydrogels with calcium for oral delivery of protein drugs,  Lin, YH; Liang, HF; Chung, CK; et al.

BIOMATERIALS  Volume: ‏ 26   Issue: ‏ 14   Pages: ‏ 2105-2113   Published: ‏ MAY 2005

[4] CONTROLLED-RELEASE OF ALBUMIN FROM CHITOSAN-ALGINATE MICROCAPSULES, POLK, A; AMSDEN, B; DEYAO, K; et al.

JOURNAL OF PHARMACEUTICAL SCIENCES  Volume: ‏ 83   Issue: ‏ 2   Pages: ‏ 178-185    1994

[5]  Light-triggered generation of multifunctional gas-filled Alginate capsules on-demand

Wang L., Wang J.Y., Song K., W. Li, Z.Huang, J. Zhu, X. Han, and Z.H. Nie,Mater. Chem. C, 2016, Volume 4, pages 652-658.

[6] Chemical, physical and biological properties of alginates and their biomedical implications

 Draget, Kurt I.; Taylor, Catherine

FOOD HYDROCOLLOIDS  Volume: ‏ 25   Issue: ‏ 2   Pages: ‏ 251-256   Published: ‏ MAR 2011

Not limited to the above representative ones, the authors should demonstrate the advantages of this work well, compared with previous reports.

2, for the experimental details, many “known amounts” or “certain amounts” appeared in the description, which should be clearly exhibited to readers for repeating the experiments.

3, The title mentioned “Localized colon-release of diclofenac sodium”, however, there is no in vivo data to prove this claim.

4, For the biodegradability, there is no clear discussion on the efficiency.

Base on the above flaws, the paper must be improved dramatically, to fulfill the high quality of this journal.

Author Response

Response: We would like to thank reviewer for the valuable comments, we have modified the manuscript and answered all comments according to the reviewer suggestions; we have improved the introduction section and provided sufficient background regarding the developed microcapsules. Experimental, results/discussion and conclusion section were also improved. All changes are highlighted in yellow.

Besides, there are major parts for revision:

1, in the introduction, the innovations are not clearly demonstrated, especially, the progress in this field is not well summarized with the existing problems. And many works on alginate microcapsules are not discussed, such as:

[1] Preparation of Ca-alginate-whey protein isolate microcapsules for protection and delivery of L. bulgaricus and L. paracasei, International Journal of Biological Macromolecules.
Volume 163, 15 November 2020, Pages 1361-1368
[2] Development, characterization and in vitro antioxidant activity of chitosan-coated alginate microcapsules entrapping Viola odorata Linn. Extract, International Journal of Biological Macromolecules, Volume 163, 15 November 2020, Pages 44-54

 [3] Physically crosslinked alginate/N,O-carboxymethyl chitosan hydrogels with calcium for oral delivery of protein drugs,  Lin, YH; Liang, HF; Chung, CK; et al. BIOMATERIALS  Volume: ‏ 26   Issue: ‏ 14   Pages: ‏ 2105-2113   Published: ‏ MAY 2005
[4] CONTROLLED-RELEASE OF ALBUMIN FROM CHITOSAN-ALGINATE MICROCAPSULES, POLK, A; AMSDEN, B; DEYAO, K; et al.
JOURNAL OF PHARMACEUTICAL SCIENCES  Volume: ‏ 83   Issue: ‏ 2   Pages: ‏ 178-185    1994
[5]  Light-triggered generation of multifunctional gas-filled Alginate capsules on-demand
Wang L., Wang J.Y., Song K., W. Li, Z.Huang, J. Zhu, X. Han, and Z.H. Nie,Mater. Chem. C, 2016, Volume 4, pages 652-658.
[6] Chemical, physical and biological properties of alginates and their biomedical implications
 Draget, Kurt I.; Taylor, Catherine. FOOD HYDROCOLLOIDS  Volume: ‏ 25   Issue: 2   Pages: 251-256.Published: ‏ MAR 2011.

Not limited to the above representative ones, the authors should demonstrate the advantages of this work well, compared with previous reports.

Response: We have improved the introduction section with more recent and related references, thanks to the reviewer for his suggestions for improving the quality of our manuscript. We have clarified the advantages of this work well, compared with previous studies.

2, for the experimental details, many “known amounts” or “certain amounts” appeared in the description, which should be clearly exhibited to readers for repeating the experiments.

Response: We have modified the experimental details and clarified the exact used quantities from samples to be clearer for readers. 

3, The title mentioned “Localized colon-release of diclofenac sodium”, however, there is no in vivo data to prove this claim.

Response: We have modified and rewritten the title of manuscript. The suggested title is "pH-Sensitive Alginate/Carboxymethyl Chitosan/Aminated Chitosan Microcapsules for Colon-Specific Release of Diclofenac Sodium"

4, For the biodegradability, there is no clear discussion on the efficiency.

Response: We have modified the discussion of biodegradability study and supported with new references. In fact, several routs have been reported for investigation the biodegradability of alginate and chitosan biopolymers. Some reports examined the biodegradation profiles via estimation the weight loss in several subsequent days (Kenawy, et al., 2019;  Lončarević, et al., 2017; Lim, et al., 2008), while others (Sun, et al., 2019; Omer, et al., 20121; Omer, et al., 2016) studies have been applied the same simple adopted method in the current study. We have performed this test as an indicator for the possible biodegradation of the developed microcapsules, which already have biodegradation nature resultant from origin nature of chitosan and alginate polysaccharides. The produced color from the reaction of the reduced liberated sugar (refers the weight loss) from microcapsules with the DNS-reagent was analyzed as a function of the optical density (OD) at a wavelength of 570 nm using a visible-spectrophotometer. It is well known that the glycosidic bonds could be hydrolyzed by the lysozyme enzyme, resulting in a biodegradation of the polysaccharide structure [65, 66]. Furthermore, the hydrophilic groups (OH, COOH and NH2) in the microcapsules matrix might be enhanced the enzymatic activity, which would potentially allow for adsorption of lysozyme enzyme. Therefore, the biodegradation rate was increased with increasing AmCs content as a result of increasing the enzyme-adsorbing groups, which positively affected the biodegradation process. These results are consistent with the authors previous works. It has been stated that microbeads, nanoparticles, and membranes based-chitosan biopolymer displayed decent biodegradability [16, 19, 38, 67].

Base on the above flaws, the paper must be improved dramatically, to fulfill the high quality of this journal.

We thank the reviewer for his valuable and important comments. We hope that the revised version of the manuscript could be suitable for publishing in Journal of Pharmaceutics

Reviewer 4 Report

Reviewer's comments:
Remarks to the Author (pharmaceutics-1067697-peer-review-v1)
Type of the Paper (Article
In the current study, Abdelazeem S. Eltaweil and co-workers developed Alg-CMCs@AmCs dual
polyelectrolyte complexes (PECs) microcapsules from carboxymethyl chitosan (CMCs), aminated
chitosan (AmCs), and alginate (Alg). The obtained microcapsules exhibited pH-triggered and
controlled drug release. Improved swelling and degradation properties were observed. However,
there are many points in the manuscript that needs to be clarified before this can be considered for
publication. The quality of this manuscript may be improved after a major revision. The specific
comments from this reviewer are given below:
1. The title of the manuscript should be modified
2. In the introduction, the authors should consider referring to the following recent
article
“Pharmaceutics 2019, 11, 621; doi:10.3390/pharmaceutics11120621”. It explored the
fabrication of a pH-responsive delivery platform for advanced drug delivery
application under gastric and intestinal pH environment.
3. The authors should add a statistical analysis section. All tables and Figures data
should include statistical analysis and point out the p-value obtained from the
statistical analysis.
4. The authors should add the FTIR and TGA curves for pure AmCs and CMCs to
Figure 2
5. The chemical structures in Figure 5 should be merged in Figure 1 to make it more
meaningful and understandable.
6. DSC curves of all the specimens should be added to the revised manuscript
7. Why the authors did select only S0, S2, S4, and S5 specimens for all the studies?
Why not all the samples reported?
8. Check the FTIR frequencies on line 213 and 214.
9. Recheck the following sentence
“1651 and1435 cm-1 (S4) as a result of second PEC layer”
10. How authors did calculate the weight loss percentage at different degradation stages
(Table 2)?
11. Spellcheck the sentence on line 320 and 356
12. There are several Figures in the manuscript, which should be reduced to a total of 5
or 6 Figures
13. Cell viability of the specimens should be measured and reported with appropriate cell
lines
14. Accordingly, the abstract and conclusions should be modified.

Author Response

  1. The title of the manuscript should be modified

Response: We ould like to thank the reviewer for his time and his important comments and suggestions. We have modified and rewritten the title of manuscript. The suggested title is "pH-Sensitive Alginate/Carboxymethyl Chitosan/Aminated Chitosan Microcapsules for Colon-Specific Release of Diclofenac Sodium"

In the introduction, the authors should consider referring to the following recent
article “Pharmaceutics 2019, 11, 621; doi:10.3390/pharmaceutics11120621”. It explored the fabrication of a pH-responsive delivery platform for advanced drug delivery
application under gastric and intestinal pH environment.

Response: We have mentioned the suggested reference in introduction section as it related to the study topic.

  1. The authors should add a statistical analysis section. All tables and Figures data
    should include statistical analysis and point out the p-value obtained from the
    statistical analysis.

Response: We have added the statistical analysis section at end of experimental section, and it included in all tables and Figures.

  1. The authors should add the FTIR and TGA curves for pure AmCs and CMCs to
    Figure 2

Response: We have add the FTIR and TGA curves for pure AmCs and CMCs to Figure 2, and the discussion was improved with relevant references.

  1. The chemical structures in Figure 5 should be merged in Figure 1 to make it more
    meaningful and understandable.

Response: We have merged Figure 5 in Figure 1.

  1. DSC curves of all the specimens should be added to the revised manuscript

Response: We have tested the thermal properties of the developed microcapsules as well as the native polymers via thermo-gravimetric analysis as presented in Figure 2b. DSC analysis is important also to get more information regarding the thermal properties.  However, according to the current conditions caused by COVID-19 virus, only one DSC-equipment is available at our organization, with a very long waiting analysis list, that could be reached up to 2 months to get the analysis results. So, we think that TGA results could be enough for identification the thermal properties, in addition to investigation the chemical structure and morphological properties by FTIR and SEM analysis tools. In addition, the cost of DSC test is expensive, since we have no fund to cover all analysis from outside, we paid from our own money. We hope that the reviewer can accept these reasons.

  1. Why the authors did select only S0, S2, S4, and S5 specimens for all the studies?
    Why not all the samples reported?

Response: As mentioned in the revised manuscript, according to our targets in this study (improve the encapsulation efficiency, improve the swelling/release profile of Alg microcapsules at colon conditions), we have selected S0, S2, S4, and S5 samples according to the optimum and best results obtained from swelling and drug encapsulation experiments (which includes all samples). Thus, characterization, the drug release experiments and biodegradability experiments were conducted by these selected samples (the pure alginate (S0; blank), one ratio of single PEC S2 [Alg-CMCs (1:1)], and two ratios of dual PECs Alg-CMCs (1:1)@ AmCs [S4 and S5].

  1. .Check the FTIR frequencies on line 213 and 214.

Response: We have checked and corrected the frequencies values

  1. Recheck the following sentence
    “1651 and1435 cm-1 (S4) as a result of second PEC layer”

Response: We have checked and corrected it to S5, and the sentence was rewritten.

  1. How authors did calculate the weight loss percentage at different degradation stages
    (Table 2)?

Response: We have calculated the weight loss percentage at different degradation stage depending on the received raw text file obtained from the TGA analysis, which describes all weight losses along with rising temperature. The text file includes also the initial weight sample and its percent (100%). Where, we have calculated the weight losses at the selected stages 25-120 oC and 220-350 oC. In addition, we have calculated the required temperature for loss the half weight (50%) of sample, and the obtained data were compared.

 Spellcheck the sentence on line 320 and 356

Response: We have checked and corrected it

  1. There are several Figures in the manuscript, which should be reduced to a total of 5
    or 6 Figures

Response: We have reduced number from 9 to 7 figures.

  1. Cell viability of the specimens should be measured and reported with appropriate cell
    lines

Response: We agree with the reviewer that cytotoxicity is very important for any developed drug delivery system. However, according to the current conditions caused by COVID-19 virus as mentioned above, it is very difficult to study cytotoxicity in this time using appropriate cell lines. In addition, we have designed our paper according to the similar published articles (using alginate and chitosan as carriers), not all of these published papers include the cytotoxicity during the in vitro studies, since the used polymers are natural biopolymers, and they have been previously tested and evaluated by authors and other studies as non-toxic materials. It has been reported that materials with toxicity < 25% are considered non-toxic.

We have previously tested the cytotoxicity of the chitosan and aminated chitosan biopolymers, and it the results clarified that toxicity level not more than 5%: as stated in the following references;

- Antioxidant and antibacterial polyelectrolyte wound dressing based on chitosan/hyaluronan/phosphatidylcholine dihydroquercetin. International Journal of Biological Macromolecules, 2021, 166, pp. 18–31.

- Enhancement of wound healing by chitosan/hyaluronan polyelectrolyte membrane loaded with glutathione: In vitro and in vivo evaluations. Journal of Biotechnology, 2020, Pages 103-113.

- Fabrication of biodegradable gelatin/chitosan/cinnamaldehyde crosslinked membranes for antibacterial wound dressing applications. International Journal of Biological Macromolecules 2019, 139, 440-448.

- Antibacterial and antioxidative activity of O -amine functionalized chitosan. Carbohydrate Polymers 169:441-450.

On the other hand, some of our work and other reported works didn’t include the Cytotoxicity as it measured previously such as:

- Formulation of Quaternized Aminated Chitosan Nanoparticles for Efficient Encapsulation and Slow Release of Curcumin. Molecules 2021, 26 (2), 449.

- Dual-layered pH-sensitive alginate/chitosan/kappa-carrageenan microbeads for colon-targeted release of 5-fluorouracil. International Journal of Biological Macromolecules. Volume 132, 1 July 2019, Pages 487-494.

-pH-sensitive ZnO/carboxymethyl cellulose/chitosan bio-nanocomposite beads for colon-specific release of 5-fluorouracil. International Journal of Biological Macromolecules. 128 (2019) 468–479.

-Preparation of magnetic pH-sensitive microcapsules with an alginate base as colon specific drug delivery systems through an entirely green route. RSC Advances 2015, 5, (6), 4628-4638.

However, cytotoxicity will be studied in the second part of this work as a future work (in vivo studies using animal test) with a chemical modification for the suggested carrier.

We hope that the reviewer can accept these reasons.

  1. Accordingly, the abstract and conclusions should be modified.

Response: We have modified the abstract and conclusion.

We thank the reviewer for his valuable and important comments. We hope that the revised version of the manuscript could be suitable for publishing in Journal of Pharmaceutics.

Round 2

Reviewer 3 Report

Thanks for the authors for the revision of the paper, however, there are still major flaws in this paper, such as the Colon-Specific Release, the main claim is not fully supported by the data. Besides, comparing the published works in this filed, also based on pH responsive materials, there is not enough contribution of this work to the field, thus difficult to fulfill the requirement of this high-level journal. 

Author Response

Response: We would like to thank the reviewer for his valuable comments; we have modified the revised manuscript to be clearer for readers, in addition to careful language revision

We have modified the title to be more eligible for the main goal of this manuscript the suggested new title is “pH-Sensitive Alginate/Carboxymethyl Chitosan/Aminated Chitosan Microcapsules for Efficient Encapsulation and Delivery of Diclofenac Sodium “. Also, abstract and conclusion were modified. More details in discussion section were added.

Contribution of this work to the field:  The interest of the researchers in alginate as a platform for the development of delivery systems has given place to a steady growth of the available literature over the last decade. As it well known that the major limitations of Alg microcapsules include their higher porosity, drug burst release at colon site and fast disintegration at colon conditions. Thus, we focused our attention in this paper to overcome the drug burst release resultant from disintegration of alginate microcapsules at the higher pH 7.2 (which similar to that in colon-site). The results were compared with our previous work and the other reported studies by other researchers (as stated in “3.5. Evaluation of DS-loading and table 3”. In addition, we have studied the swelling and drug release profiles in simulated GI-TRACT (in vitro studies).The results were also discussed from the view of literature.

- Comparing with other reported studies, the obtained results proved that mixing of alginate with CMCs and followed by coating with AmCs derivative clearly prevent the burst release process for DS-drug, enriched the drug loading efficiency to a maximum value of 95.4% compared to native Alg (only 68.6%). The swelling and release time were significantly extended up to 8h (similar to the overall transit time in GI-tract) compared to 3 h (for Alg). Besides, the results signify that AmCs concentration plays an important role to get sustained release profiles. Besides we have investigated the biodegradability study. The second part of this work will be oriented to study in details the in vivo drug release profiles as well as investigate the biological properties of the developed carriers.

- Similar to the design of this paper, several published papers (examples below) have been designed by the authors and other researchers in literature for development of microbeads based-alginate, and focused their work to solve one problem not all drawbacks. Some of these papers have been published also without in vivo studies.

- We hope that the revised version of the manuscript could be suitable for publishing in Journal of Pharmaceutics.

-Omer, A. M.; Ziora, Z. M.; Tamer, T. M.; Khalifa, R. E.; Hassan, M. A.; Mohy-Eldin, M. S.; Blaskovich, M. A., Formulation of Quaternized Aminated Chitosan Nanoparticles for Efficient Encapsulation and Slow Release of Curcumin. Molecules 2021, 26 (2), 449.

-Sun, X.; Liu, C.; Omer, A.M.; Lu, W.; Zhang, S.; Jiang, X.; Wu, H.; Yu, D.; Ouyang, X.K. Ph-sensitive Zno/carboxymethyl cellulose/chitosan bio-nanocomposite beads for colon-specific release of 5-fluorouracil.Int. J. Biol. Macromol. 2019, 128, 468–479.

-Sun, X.; Liu, C.; Omer, A.M.; Yang, L.Y.; Ouyang, X.K. Dual-layered ph-sensitive alginate/chitosan/kappacarrageenan microbeads for colon-targeted release of 5-fluorouracil. Int. J. Biol. Macromol. 2019, 132, 487–494.

Reviewer 4 Report

All my concerns are well addressed, and now it can be published.

Congratulation!

Author Response

Response, We would like to thank the reviewer for his time for reviewing of our manuscript, we believe that these comments regarding the original manuscript have helped us to raise the paper quality.

Best regards

Round 3

Reviewer 3 Report

Thanks for the authors for their careful and detailed response. By comparing the very recent publications, including the ones listed by the authors, the referee believes that the paper could be definitely published eventually, however, probably not in the journal of Pharmaceutics, due to the limited contribution in the field of pharmaceutics.

The reviewer would be happy to suggest the Editor to transfer this paper to a sister journal of MDPI.

Author Response

Dear Editor:

Thanks for the reviewer for his response. We have corrected and modified the the manuscript as much as we can, according to all reviewers comments, and we have checked the language throughout the whole manuscript.  We agree with the reviewer that the experiments regarding the field of pharmaceutics is not more enough, however, we hope that the studied swelling and release profiles under the simulated gastrointestinal conditions could present some information for readers regarding improvement the drug encapsulation efficacy of Alg microcapsules, and overcome their drawbacks at simulated colon pH.  On the other hand, an reported study have been published by the journal pharmaceutics, an the authors have been studied few only one release experiment (at pH1.2 AND pH 7.4). 

-Sun et al. Fabrication of Ion-Crosslinking Aminochitosan Nanoparticles for Encapsulation and Slow Release of Curcumin. Pharmaceutics. 2019;11(11):584.  doi:10.3390/pharmaceutics11110584